# Extracellular Nucleotides Regulate Arterial Calcification by Activating Both Independent and Dependent Purinergic Receptor Signaling Pathways

**DOI:** 10.3390/ijms21207636

**Published:** 2020-10-15

**Authors:** Britt Opdebeeck, Isabel R. Orriss, Ellen Neven, Patrick C. D’Haese, Anja Verhulst

**Affiliations:** 1Laboratory of Pathophysiology, Department of Biomedical Sciences, University of Antwerp, 2000 Antwerpen, Belgium; britt.opdebeeck2@uantwerpen.be (B.O.); ellen.neven@uantwerpen.be (E.N.); anja.verhulst@uantwerpen.be (A.V.); 2Department of Comparative Biomedical Sciences, Royal Veterinary College, London NW1 0TU, UK; iorriss@rvc.ac.uk

**Keywords:** arterial calcification, purinergic signaling, pyrophosphate, ecto-nucleotidases, extracellular nucleotides

## Abstract

Arterial calcification, the deposition of calcium-phosphate crystals in the extracellular matrix, resembles physiological bone mineralization. It is well-known that extracellular nucleotides regulate bone homeostasis raising an emerging interest in the role of these molecules on arterial calcification. The purinergic independent pathway involves the enzymes ecto-nucleotide pyrophosphatase/phosphodiesterases (NPPs), ecto-nucleoside triphosphate diphosphohydrolases (NTPDases), 5′-nucleotidase and alkaline phosphatase. These regulate the production and breakdown of the calcification inhibitor—pyrophosphate and the calcification stimulator—inorganic phosphate, from extracellular nucleotides. Maintaining ecto-nucleotidase activities in a well-defined range is indispensable as enzymatic hyper- and hypo-expression has been linked to arterial calcification. The purinergic signaling dependent pathway focusses on the activation of purinergic receptors (P1, P2X and P2Y) by extracellular nucleotides. These receptors influence arterial calcification by interfering with the key molecular mechanisms underlying this pathology, including the osteogenic switch and apoptosis of vascular cells and possibly, by favoring the phenotypic switch of vascular cells towards an adipogenic phenotype, a recent, novel hypothesis explaining the systemic prevention of arterial calcification. Selective compounds influencing the activity of ecto-nucleotidases and purinergic receptors, have recently been developed to treat arterial calcification. However, adverse side-effects on bone mineralization are possible as these compounds reasonably could interfere with physiological bone mineralization.

## 1. Introduction

Our genetic code has a well-defined arrangement of building blocks or nucleotides adenosine 5′-triphosphate (ATP), uridine 5′-triphosphate (UTP), cytidine 5′-triphosphate (CTP) and guanosine 5′-triphosphate (GTP) which are translated into proteins. Interestingly, nucleotides are not only located intracellularly and cells indeed have the capacity to release nucleotides into the extracellular space through either a controlled release mechanism or via cell death. In the 1970s, Geoffrey Burnstock discovered that extracellular nucleotides play an important role as signaling molecules [1]. Extensive research has demonstrated that extracellular nucleotides, mainly ATP and to a lesser extent UTP and adenosine-5′-diphosphate (ADP), regulate many biological processes including blood coagulation, wound healing, inflammation, cancer and bone formation [2,3,4]. Normally, the basal concentration of ATP intracellularly is around 3–5 mM while outside the cell the concentration of ATP should be around 3–500 nM to activate nearby purinergic receptors. Furthermore, extracellular nucleotides have a short half-life (measured in seconds) due to the close proximity of ecto-nucleotidases or enzymes which are involved in the breakdown of extracellular nucleotides. This makes that their paracrine radius is only a few hundred microns [5]. Numerous studies have shown that extracellular nucleotides influence bone remodeling, acknowledged by multiple reviews [6,7,8]. Interestingly, due to the striking similarities between physiological bone formation and pathological arterial calcification, there is an emerging interest in the role of extracellular nucleotides in pathological mineralization in the vessel wall which will be discussed in detail in the next paragraphs.

## 2. Arterial Calcification

Arterial calcification, or the accumulation of calcium-phosphate crystals (i.e., hydroxyapatite) in the arterial wall or valves, has a significant impact on morbidity and mortality in chronic kidney disease (CKD), osteoporosis and diabetes patients by provoking severe cardiovascular events [9]. Four types of arterial calcification can be distinguished according to their location being (i) calcification in the intimal layer of the arterial wall in association with atherosclerotic plaques or arterial intima calcification, (ii) calcification in the medial layer of the arterial wall, called arterial media calcification or Monckeberg’s arteriosclerosis, (iii) aortic valve calcification and (iv) calciphylaxis or calcific uremic arteriolopathy or calcification of the small blood vessels in the skin. Each subtype of arterial calcification has its own specific risk factors and outcomes, shown in Figure 1 adapted from [10,11,12].

Furthermore, cardiac calcification at the mitral annulus is a process possessing characteristics of both atherosclerotic plaque calcification and media calcification [13]. In general, the process of arterial calcification has three important pathological characteristics (i) loss of circulating calcification inhibitors (i.e., pyrophosphate (PP_i_), (vitamin K-dependent) carboxylated matrix gla protein and fetuin-A) [14,15], (ii) gain of circulating calcification stimulators (i.e., hyperphosphatemia, hypercalcemia and uremic toxins) [15,16] and (iii) the ability of vascular cells, i.e., vascular smooth muscle cells (VSMCs) in the (medial or intimal) arterial wall or valve interstitial cells (VICs) in aortic valves, to transdifferentiate into bone-like cells which goes along with upregulation of osteo/chondrogenic marker genes such as Runt-related transcription factor 2 (Runx2), tissue non-specific alkaline phosphatase (TNAP), Bone morphogenetic protein 2 (BMP2) and SRY-Box transcription factor 9 (Sox9). These bone-like cells produce and release calcified matrix vesicles, in which calcium and phosphate is built up, into the extracellular matrix resulting into mineralization of the surrounding tissue [15,16]. Besides VSMCs and VICs, endothelial cells also play a crucial role in the process of arterial calcification by (i) the release of nitric oxide (NO) and (ii) undergoing endothelial to mesenchymal transition. Results from in vitro and in vivo studies form an argument for the fact that NO inhibits VSMC calcification and osteo/chondrogenic transdifferentiation. Endothelial to mesenchymal transition contributes to arterial calcification by the generation of mesenchymal stem-like cells that can eventually migrate to the medial layer and differentiate further into multiple cell lineages: fibroblasts/myofibroblasts, osteoblasts/osteocytes and chondrocytes [17].

Extracellular nucleotides, such as ATP and UTP, potently inhibit this extracellular matrix mineralization via dual inhibitory pathways (Figure 2): (i) via their hydrolysis product PP_i_, which binds to hydroxyapatite crystals and prevents further incorporation of inorganic phosphate (P_i_) into the crystals (purinergic receptor independent pathway) and (ii) by directly activating purinergic receptors (purinergic receptor dependent pathway). Furthermore, ATP and UTP are involved in purinergic receptor signaling (purinergic receptor dependent pathway) and are hydrolyzed by ecto-nucleotidases to generate PPi (purinergic receptor independent pathway), while the extracellular nucleotides CTP and GTP are only involved in the purinergic receptor independent pathway, as they do not interact with purinergic receptors [18]. This review will discuss the role of both the purinergic receptor independent and dependent pathways in arterial calcification.

## 3. Purinergic Receptor Independent Pathway

PP_i_ is a key player in the purinergic receptor independent pathway as it is one of the most powerful endogenous calcification inhibitors which binds to nascent hydroxyapatite crystals thereby preventing further incorporation of P_i_ into these crystals [19]. The family of ecto-nucleotidases play an indispensable role in regulating the production and breakdown of PP_i_ and P_i_, both controlling the mineralization processes, from extracellular nucleotides. Four families of ecto-nucleotidases are distinguished, being: (1) ecto-nucleotide pyrophosphatase/phosphodiesterases (NPPs); (2) ecto-nucleoside triphosphate diphosphohydrolases (NTPDases); (3) 5′-nucleotidase, and (4) alkaline phosphatase [20]. Maintaining a proper plasma PP_i_/P_i_ ratio is critical as (i) lower plasma PP_i_ in e.g., dialysis patients negatively correlates with arterial calcification [21], (ii) multiple animal models have shown that an increase in plasma PP_i_ inhibits arterial calcification [22,23,24,25] and (iii) oral administration of PP_i_ to healthy volunteers recently resulted in a transient elevation of plasma PP_i_, indicating that oral PP_i_ supplementation might be a promising therapy for the treatment of arterial calcification in particular patient population such as infants who suffer from arterial calcification and low levels of plasma PP_i_, a disorder called generalized arterial calcification of infancy [26]. The next paragraphs will discuss the involvement of the different ecto-nucleotidases, controlling the plasma PP_i_/P_i_ ratio, in the calcification process of the arterial wall.

### 3.1. Involvement of NPPs in Arterial Calcification

NPPs are plasma membrane proteins known to hydrolyze a wide range of phosphodiester bonds, e.g., nucleoside triphosphates, (cyclic) dinucleotides, and nucleotide sugars [20]. Seven subtypes of NPPs (NPP1–7) exist whereby NPP1 and NPP3 are suggested to play a role in the arterial calcification process. NPP1 stimulates the hydrolysis of ATP into AMP and PP_i_ however it has also been implicated, to a lesser extent, in generating P_i_ [27]. Generalized arterial calcification of infancy, an autosomal-recessive genetic disorder, causes the premature onset of arterial calcification due to a functional absence of the NPP1-encoding gene (ENPP1) leading to a decrease in NPP1 activity and thus a reduction in the generation of PP_i_. This was also confirmed in an ENPP1 knockout mouse model that mimics this disease phenotype and leads to the development of arterial calcification [28]. Moreover, NPP1 enzyme replacement therapy prevented the development of arterial calcifications in a mouse model of generalized arterial calcification of infancy [29]. Similarly, pseudoxanthoma elasticum, which is characterized by the development of ectopic calcification of soft connective tissues, is associated with ENPP1 missense variants [30]. Alternatively, an overexpression of NPP1 activity is also linked to ectopic calcification because it (i) triggers TNAP activity (discussed in detail below) to produce P_i_ in order to restore the PP_i_/P_i_ ratio and (ii) depletes the ATP pool leading to a reduction in purinergic receptor signaling (discussed in detail below) [31]. NPP1 inhibitors have been developed as potential therapeutics for diabetes mellitus and brain cancers [32]. Additionally, a recent in vitro study showed that treating mineralized VICs—which overexpress NPP1—with a specific non-competitive NPP1 inhibitor led to a significant (*p* < 0.05) decrease in mineral content, apoptosis and osteo/chondrogenic transdifferentiation, as compared to VICs on a pro-calcifying medium without inhibitor supplementation [33]. In conclusion, it is key to maintain the NPP1 activity in a well-defined range as overexpression, as well as reduced expression of NPP1 activity, have been linked to arterial calcification. Furthermore, NPP3, an enzyme that is also known as basophil-specific ecto-enzyme E-NPP3 (CD203c) is involved in the allergic inflammation response. Basophils are activated by the binding of an antigen to an immunoglobulin E, favoring the release of inflammatory mediators and upregulation of NPP3 to the cell surface. Subsequently, NPP3 upregulation induces hydrolysis of extracellular ATP, a pro-inflammatory mediator, leading to the suppression of chronic allergic inflammation [34]. With regard to its role in arterial calcification, a study by Villa-Bellosta et al. suggests that in VSMCs, NPP3 also participates in PP_i_ hydrolysis [35].

In conclusion, both absence/reduced presence and overexpression of NPP1 can induce arterial calcification, implying that keeping its activity in a well-defined molecular range is crucial. Furthermore, the role of NPP3 in the arterial calcification process has to be further investigated.

### 3.2. Involvement of Alkaline Phosphatase in Arterial Calcification

Four types of alkaline phosphatase exist including three tissue-specific isozymes intestinal, placental and germ-cell alkaline phosphatase and TNAP [20]. In human plasma, 95% of the alkaline phosphatase activity is attributed to the TNAP isozyme which is mainly expressed by the liver, kidney and bone [36]. In the liver and kidney, TNAP plays a pivotal role in anti-inflammatory actions through dephosphorylation of the bacterial endotoxin lipopolysaccharide and probably by depletion of the ATP pool, released during cell stress, as ATP attracts phagocytes and platelets and activates the nucleotide-binding leucine-rich repeat (NLR) family pyrin domain containing 3 (NLRP3) inflammasome [37]. While in the bone, TNAP is produced by osteoblasts to maintain adequate bone mineralization by the degradation of PP_i_ into P_i_ [7]. Similar to this, VSMCs are capable of expressing TNAP under osteogenic circumstances to promote the calcification process [15]. TNAP is loaded into calcified matrix vesicles under control of the sorting receptor sortilin, thereby favoring the aggregation/accumulation of calcium-phosphate crystals [38]. A recent study has suggested that TNAP is potentially cleaved from the calcified matrix vesicles just before binding to the extracellular matrix as TNAP disturbs interactions between annexin a5, a collagen-binding protein present in the calcified matrix vesicles, and the extracellular matrix [39]. Furthermore, endoplasmic reticulum stress induced during arterial calcification, regulates alkaline phosphatase mRNA production and activity in VSMCs by interacting with the activating transcription factor 4 (ATF4) [40]. Interestingly, apabetalone, a recently introduced novel inhibitor of bromodomain and extraterminal (BET) proteins which binds to transcription factors to regulate gene expression, has been suggested to disrupt the interaction between BET protein 4 and an activating transcription factor 3 (ATF3) [41]. Moreover, apabetalone decreased major adverse cardiac events in cardiovascular disease patients [42] and significantly (*p* < 0.02) diminished circulating alkaline phosphatase levels in CKD patients versus CKD patients treated with a placebo [43]. Additionally, apabetalone blocked transdifferentiation and calcification of VSMCs through halting the TNAP gene expression, protein levels and enzyme activity [41].

In hemodialysis patients, serum TNAP levels have been associated with significantly increased coronary artery calcium scores (OR 3.89, 95% CI (2.01; 7.54), *p* = 0.001) and abdominal aortic calcification (r = 0.389; *p* < 0.01) [44,45]. Moreover, transgenic mouse models in which TNAP was selectively over-expressed in either VSMCs or endothelial cells led to the development of arterial media calcifications [46,47]. On the contrary, low levels of TNAP activity induce a state of hypophosphatasia, characterized by fractures, arthralgia and neurological complications [48]. Consequently, it is essential to maintain the level of TNAP activity in a well-defined molecular range, also observed for NPP1 activity (see above). In this respect, investigating the cut-off values of the ecto-nucleotidase enzyme activities in plasma/serum could be interesting to predict adverse outcomes including arterial calcification. In conclusion, influencing the activity of TNAP would be an eligible therapeutic approach for arterial calcification and as such will stimulate the discovery of selective and potent TNAP inhibitors. The first inhibitors were non-selective inhibitors of alkaline phosphatase, acting through uncompetitive or competitive interactions with the enzyme. For example, L-phenylalanine, imidazole, histamine, and theophylline are uncompetitive inhibitors whilst phosphate, phosphoethanolamine, and phenylphosphonate inhibit alkaline phosphatase competitively [49]. Later on, selective, uncompetitive inhibitors against TNAP were developed which can be classified into three groups (i) series of sulfonamide inhibitors [50], (ii) series of triazolyl pyrazole inhibitors [51] and (iii) series of coumarin sulphonate inhibitors [52]. Multiple studies have proven that the selective TNAP inhibitor 5-((5-chloro-2-methoxyphenyl)sulfonamido)nicotinamide (SBI-425) is an effective therapy against arterial calcification in multiple mouse and rat models for arterial media calcification and pseudoxanthoma elasticum [46,47,53,54,55]. With regard to physiological bone mineralization, treatment with SBI-425 went along with a decreased bone formation rate and increased osteoid maturation time in a rat model with warfarin-induced arterial calcification [55] and which was not seen in previous mouse studies [46,53,54]. For this reason, investigating the therapeutic effect of SBI-425 in other species, large skeletally mature animals including dogs and rabbits who have a bone structure highly similar to that of humans, is indispensable [56], in particular given the knowledge that CKD patients, who are a major target population for treating arterial calcification, already suffer from bone metabolic defects. Lastly, orphan phosphatase 1 (PHOSPHO1) relates closely to the family of alkaline phosphatases. PHOSPHO1 initiates the hydrolysis of intravesicular metabolites including phosphoethanolamine and phosphocholine to obtain P_i_. Thus, it has been suggested that PHOSPHO1 functions synergistically with TNAP to accumulate P_i_ into the calcified matrix vesicles [57]. Moreover supplementation of a PHOSPHO1 inhibitor alone and in combination with a TNAP inhibitor, prevented in vitro VSMC calcification [58].

In conclusion, targeting TNAP activity to treat arterial calcification has received a lot of interest. However, similarly to NPP1 activity, monitoring TNAP activity in the circulation is necessary as too high levels of TNAP activity correlate to the development of arterial calcification while too low levels associate with bone mineralization defects and neurological defects.

### 3.3. Involvement of NTPDases and 5′-Nucleotidase in Arterial Calcification

The NTPDases or CD39 superfamily contains eight members including the cell surface NTPDase 1–3,8 and the intracellular NTPDase 4–7 which catalyze the hydrolysis of ATP/ADP into ADP/AMP under the formation of P_i_. Furthermore, 5′-nucleotidase or CD73 produces adenosine by the hydrolysis of AMP [59]. Studies have shown that the activity of NTPDase 1 and 5′-nucleotidase was significantly (*p* < 0.05) lower in calcified aortic valves versus non-calcified aortic valves suggesting a potential role of both ecto-nucleotidases in aortic valve calcification [60,61]. Moreover, ARL67156, an NTPDase inhibitor, however also a non-specific purinergic P2 receptor inhibitor, prevented the development of aortic stenosis and calcification of the aorta and aortic valves in a warfarin rat model [62]. In addition, a recent study showed that a combination therapy of ATP, the TNAP inhibitor—levamisole, and NTPDase inhibitor—ARL67156, reduced arterial calcification and extended longevity in a Hutchinson–Gilford progeria syndrome mouse model, which was accomplished by increasing the PP_i_ production through preventing (i) TNAP mediated hydrolysis of PP_i_ into P_i_ and (ii) NTPDase mediated reduction in the availability of extracellular ATP for PP_i_ production by NPPs. Hutchinson–Gilford progeria syndrome is a rare genetic disorder observed in children who suffer from premature aging and cardiovascular disease including vascular stenosis and excessive arterial calcification [63]. Until today, little information on the role of the different subtypes of NTPDases in arterial calcification exists, probably due to a lack of specific synthetic NTPDase inhibitors. A rare disease called arterial calcifications due to deficiency of 5′-nucleotidase/CD73 (ACDC), caused by autosomal recessive mutations in the CD73-encoding gene (NT5E), is characterized by painful calcification in the arteries and joints of the lower extremities [64]. Interestingly, osteogenic stimulation of induced pluripotent stem cell-derived mesenchymal stromal cells from ACDC patients caused significantly (*p* < 0.001, after 21 days versus control human dermal fibroblasts) higher levels of TNAP resulting in lower levels of PP_i_ and activation of the Akt/mammalian target of rapamycin (mTOR) pathway—an arterial calcification stimulating pathway [65]. Additionally, a study using primary dermal fibroblasts obtained from ACDC patients demonstrated that the absence of 5′-nucleotidase activity induced a decrease in adenosine production. The subsequent reduction in adenosine-mediated purinergic P1 receptor signaling and intracellular cyclic adenosine monophosphate (cAMP), led to activation of the Akt/mTOR pathway. Subsequently, phosphorylated Akt stimulates the nuclear translocation of the transcription factor forkhead box O1 protein (FOXO1) to promote the expression and activity of TNAP and thus stimulates the arterial calcification process [66]. Additionally, adenosine derived from 5′-nucleotidase activity influences the arterial calcification process by interacting with purinergic P1 receptors, which will also be discussed in more detail below. In conclusion, the different types of ecto-nucleotidases are closely entwined in the arterial calcification process shown in Figure 3.

## 4. Purinergic Receptor Dependent Pathway

Once nucleotides and nucleosides are released in the extracellular space, they have the ability to bind to purinergic receptors. Two broad family types of purinergic receptors exist, being P1 and P2 receptors. The G protein-coupled P1 receptor family contains four subtypes A_1_, A_2A_, A_2B_ and A_3_ which are activated by adenosine. The P2 receptor family can be further subdivided into eight G protein-coupled P2Y receptors (P2Y_1,2,4,6,11,12,13,14_) and seven ligand-gated ion channels, the P2X receptors (P2X1–7) [67,68]. Purinergic signaling plays an important role in regulating both osteoblast mediated mineralization and osteoclast mediated resorption of the bone matrix [3,18,69]. As already mentioned, the pathological process of arterial calcification mimics physiological bone mineralization. Consequently, the ability of purinergic receptors to influence ectopic mineralization has become an area of interest. The research group of Schuchardt et al. was one of the first to provide evidence for the involvement of P2 receptors in the arterial calcification process. Upon shear stress and hypoxia, endothelial cells release the contracting factor uridine adenosine tetraphosphate (Up_4_A) which stimulates the P2X and P2Y receptors on VSMCs, leading to a phenotypic switch into osteoblast-like cells with upregulation of alkaline phosphatase and Runx2 [70]. However Up_4_A is not a selective P2 receptor agonist and only weakly activates some P2 receptors, therefore results need to be interpreted with caution. Patel. et al. showed that the inhibitory effect on VSMC calcification mediated by ATP and UTP is not solely attributed to its breakdown product PP_i_ and also involves P2 receptor activation [71]. The mRNA expression of the P2X1, P2X2, P2X4, P2X5, P2X6, and P2Y_2_ receptors was increased by up to three times in calcifying VSMCs compared to control VSMCs suggesting a potential role in ectopic mineralization [71]. The next paragraphs will discuss in depth the involvement of P1 and P2 receptors in arterial calcification.

### 4.1. Involvement of P1 Receptors in Arterial Calcification

Adenosine, the endogenous ligand of P1 receptors (A_1_, A_2A_, A_2B_ and A_3_), induces opposing effects depending on the tissue and activation of its receptor subtype. In general, the activation of A_1_ and A_3_ receptor subtypes reduce intracellular cAMP, whilst triggering theA_2A_ and A_2B_ receptor subtypes induces the opposite effect [72]. Elevated cAMP signaling and high P_i_ levels synergistically stimulate in vitro VSMC calcification [73]. In line with these results, activation of the A_2A_ receptor promoted the mineralization process in VIC cultures by favoring the cAMP/protein kinase A pathway, while activation of the A_1_ receptor resulting in decreased cAMP levels, inhibited VIC mineralization [31]. Moreover, adenosine treatment of mice aortic roots exposed to an osteogenic medium attenuated mineralization by A_1_ and A_2B_ receptor activation, whereas the A_2A_ receptor had an opposite effect on aortic root mineralization [74]. Recently, an in vitro study using human aortic smooth muscle cells demonstrated that triggering the A_3_ receptor resulted in a reduction in matrix mineralization, probably by decreasing alkaline phosphatase activity [75]. Furthermore, adenosine also plays a role in the lineage-specific differentiation of mesenchymal stem cells into osteoblasts and adipocytes mediated by upregulation of the A_2A_ receptor and A_1_/A_2B_ receptor, respectively [76]. Interestingly, adipocyte-related protein peroxisome proliferator-activated receptor-gamma (PPARgamma) blocks VSMC calcification by suppressing Wnt/β-catenin signaling mediated osteo/chondrogenic transdifferentiation [77,78,79]. This raises the idea that favoring the VSMC phenotypic switch towards an adipogenic phenotype instead of an osteoblastic phenotype, might be beneficial to counteract arterial calcification. However, more extensive research is needed to confirm this theory. Furthermore, the process of endothelial cells undergoing an endothelial to mesenchymal transition, a subtype of epithelial to mesenchymal transition, has been associated with arterial calcification in mice lacking the matrix gla protein [80]. Interestingly, activation of both P1 and P2 receptors (discussed below) play a significant role in the epithelial to mesenchymal transition in different pathologies [81]. However, investigating the role of purinergic receptor signaling in endothelial to mesenchymal transition in endothelial cells and arterial calcification, would be a novel and interesting route to explore. Taken together, the effect of adenosine on the arterial calcification process depends on the activated P1 receptor subtype and thus limits the therapeutic use of adenosine for treating arterial media calcification. Developing synthetic selective P1 receptor subtype agonist/antagonists, however, might be a strategy to tackle the arterial calcification process. Figure 4 gives an overview of the different P1 receptors in the arterial calcification process.

### 4.2. Involvement of the P2Y Receptors in Arterial Calcification

The P2Y_2_ receptor is the most extensively studied purinergic receptor in the context of arterial calcification. Coté et al. showed that treatment of calcified VICs with a selective P2Y_2_ receptor agonist, 2-thioUTP, resulted into a reduction in mineralization which was associated with the activation of the phosphoinositide 3-kinase (PI3K) signaling pathways preventing VIC apoptosis [82]. In agreement, another study demonstrated that stimulating the P2Y_2_ receptor in VICs with 2-thioUTP induced the PI3K signaling pathway. This inhibited NFkB signaling and led to the suppression of IL-6 expression, a calcification stimulator which acts by its ability to upregulate the osteogenic transcripts Runx2 and BMP2 [83]. Furthermore, these results were reinforced by an in vivo study where mice already suffering from calcific aortic valve stenosis were treated with intraperitoneal injections of saline or 2 mg/kg body weight/day 2-thioUTP. Interestingly, researchers observed a significant improvement in left ventricular function (*p* = 0.0002 versus saline treatment), less left ventricular hypertrophy (*p* = 0.02 versus saline treatment) and a decrease in aortic valve mineral content (*p* = 0.006 versus saline treatment) after two months of 2-thioUTP treatment, without side-effects on the bone mineral content. The 2-thioUTP mediated regression of calcification of the aortic valve was attributed to acidification of the extracellular space by upregulation of carbonic anhydrase [84]. These results were very exciting as regression of ectopic calcification is rarely observed, certainly, without inducing deleterious effects on bone metabolism. Furthermore, apolipoprotein E knockout mice, susceptible to the development of arterial intima calcification, with a P2Y_2_ receptor knockout background developed significantly (*p* < 0.001) more extensive calcification in the intima layer as compared to apolipoprotein E knockout mice expressing the P2Y_2_ receptor [85]. In contrast, an in vitro study using cells from P2Y_2_ knockout mice, found no involvement of the P2Y_2_ receptor in the calcification of VSMCs [71], the key players in arterial media calcification and intimal atherosclerotic plaque calcification. This suggests that different mechanisms are in place according to the location of calcifications in the arterial tree.

Arterial media calcification is associated with arterial stiffening. A vicious cycle of arterial stiffness and arterial media calcification is described in which mechano-sensing plays an important role as VSMCs sense matrix stiffness, which in turn induces alterations in gene expression and thus, stimulates the phenotypic conversion of VSMCs [17]. Interestingly, activation of the P2Y_2_ receptor has been associated with mechano-sensing in osteoblasts and chondrocytes [86,87,88]. During mechanical loading, stimulating the bending of bones and matrix deformation, a pressure gradient is built up in the interstitial fluid. Osteocytes are connected by narrow channels, canaliculi, through which the interstitial fluid has to be squeezed, creating a high wall shear stress within these channels [89]. This fluid shear stress triggers an extracellular calcium influx, by activation of mechanosensitive and voltage-sensitive calcium channels, leading to extracellular release of ATP and UTP from vesicles stored in the osteoblast. Both nucleotides activate the P2Y_2_ receptor mediated RhoA/Rock pathway resulting in the formation of actin stress fibers and thus modifying osteoblast stiffness [86]. Similarly to this, vascular wall shear stress initiates the release of ATP by endothelial cells stimulating the P2Y_2_ receptor which is followed by the activation of endothelial nitric oxide synthase (eNOS) [90]. Additionally, another study indicated that P2Y_1_, P2Y_2_ and potentially P2Y_4_ receptors activate phospholipase C leading to the production of inositol trisphosphate (IP3) and diacylglycerol, favoring protein kinase C activation and the release of intracellular calcium stores, ultimately leading to the phosphorylation/activation of eNOS in endothelial cells [91]. NO induces vasodilatation through relaxation of the VSMCs. Several studies have shown that NO limits the development of arterial calcification [92,93,94], indicating that the interplay between endothelial cells and VSMCs, in part through the P2Y_2_ receptor, is important to halt the calcification process in the arterial wall. Taken together, studies have demonstrated that triggering the P2Y_2_ receptor inhibits intima and aortic valve calcification (i) directly by acting on the vascular cell through halting apoptosis, expressing IL6 and stimulating the resorption of minerals by an acidification process, (ii) indirectly by probably stimulating endothelial cell mediated NO production. The role of the P2Y_2_ receptor in arterial media calcification remains inconclusive and demands further research.

At present, information about the involvement of other P2Y receptor subtypes in the arterial calcification process is limited. Studies have shown that the P2Y_1_, P2Y_4_, P2Y_6_ and P2Y_12_ receptors are involved in the regulation of bone cell function. For example, overexpression of the P2Y_1_ receptor in human spinal ligament cells induced mineralization and favored the expression of osteogenic marker genes BMP2 and Sox9 which was obviated by a selective P2Y_1_ receptor antagonist [95]. Furthermore, UDP mediated P2Y_6_ receptor activation triggered the formation of osteoclasts from precursor cells [96]. Therefore one may assume that engagement of the P2Y_6_ receptor promotes the transdifferentiation of VSMCs into bone-resorbing cells and by this method, reverses the calcification process in the arterial wall. Several studies showed the possibility of reversibility of arterial calcification [97,98,99]. Lastly, clopidogrel, a selective P2Y_12_ receptor antagonist, halted mineralized bone nodule formation, alkaline phosphatase activity and collagen formation while stimulating adipocyte formation in osteoblast cell cultures [100]. Additionally, adipogenic differentiation of human bone marrow-derived mesenchymal stem cells was stimulated by the activation of the P2Y_1_ and P2Y_4_ receptor [101]. Knowing this, it will be interesting to evaluate whether the P2Y_1_, P2Y_4_ and P2Y_12_ receptors might regulate VSMCs transdifferentiation into an adipogenic phenotype (as discussed in the paragraph above dealing with the P1 receptors) to prevent arterial calcification. As such, in vitro and subsequently in vivo evaluation of these alternative P2Y receptors on arterial calcification would be of interest to establish novel, innovative treatments for arterial calcification. Although, an important remark is that P2 receptors have overlapping specificity, i.e., the P2Y_4_ receptor has an overlapping agonist profile with the P2Y_2_ receptor. Therefore it is likely that there is a degree of redundancy in the purinergic system, thus inhibiting a specific P2 receptor may stimulate another P2 receptor to compensate for its loss of function. Figure 5 shows an overview of the involvement of P2Y receptors in arterial calcification.

### 4.3. Involvement of the P2X Receptors in Arterial Calcification

With regard to the P2X receptors, research is predominantly focused on the role of the P2X7 receptor in the process of mineralization. However, its exact function remains inconclusive as published studies are conflicting. Multiple reports have shown that activation of the P2X7 receptor favors the osteogenic differentiation of multiple cell types including human mesenchymal stem cells [102], rat calvarial cells [103], human osteosarcoma cancer cells [104] and bone marrow-derived mesenchymal stem cells [105,106]. Additionally, activation of the P2X7 receptor on endothelial cells enhances apoptotic events [107] and thus, might stimulate the arterial calcification process as apoptotic bodies are attractive nucleation sites for calcium-phosphate crystals. However, Orriss et al. demonstrated that exposing osteoblasts to a P2X7 receptor agonist, Bz-ATP, or P2X1 and P2X3 receptor agonists, α,β-meATP and β,γ-meATP, significantly inhibited bone mineralization (*p* < 0.001 versus control ((without agonist supplementation) osteoblast) and alkaline phosphatase activity (*p* < 0.05 versus control ((without agonist supplementation) osteoblast) [108]. Moreover, a recent study has shown that these agonists inhibit VSMC calcification in vitro by diminishing VSMC apoptosis and osteo/chondrogenic switch. Nevertheless these observed inhibitory effects on VSMC calcification were not prevented by the use of selective P2X receptor antagonists, PPADS (non-selective), NF110 (P2X3), 5-BDBD/PSB-12062 (P2X4) and A740003/AZ10606120 (P2X7), indicating the activation of P2X receptor-independent mechanisms [109]. Hence, the exact mechanism by which Bz-ATP, α,β-meATP and β,γ-meATP inhibit VSMC calcification remains unclear. So far, nobody else has investigated the involvement of P2X receptors in arterial calcification. Interestingly, high levels of extracellular magnesium halt matrix mineralization of tendon-derived stem cells by suppressing the expression of P2X4, P2X5 and P2X7 receptor and activating the expression of the P2Y_1_, P2Y_2_, P2Y_4_ and P2Y_14_ receptors [110]. Several studies demonstrated a relationship between dietary magnesium and the prevention of arterial calcification through inhibiting osteo/chondrogenic transdifferentiation of VSMCs and stopping the maturation process of amorphous calcium-phosphate particles into stable hydroxyapatite crystals [111]. Taken together, magnesium might also inhibit arterial calcification by regulating the above mentioned P2 receptors in the arterial wall, however this needs to be further investigated.

In conclusion, purinergic signaling, as well as ecto-nucleotidases play an indispensable role in the arterial calcification process. Novel therapeutic approaches were developed to treat arterial calcification including selective TNAP and NPP1 inhibitors as well as synthetic ATP analogues (i.e., 2-thioUTP, Bz-ATP, α,β-meATP and β,γ-meATP). Based on the recent literature, these treatments have promising beneficial outcomes on arterial calcification. However, the role of purinergic signaling in the regulation of these processes is by no means fully defined and further in vitro and in vivo studies are warranted to elucidate the mechanisms involved. Moreover, awareness for possible side-effects at the level of the bone is imperative, as possible interference of these compounds with physiological bone mineralization cannot be excluded. This consideration is of particular importance to CKD patients, a major target group for treatment of arterial calcification which inherent to their disease state, already suffer from compromised bone metabolism. Having this in mind, it needs to be emphasized that in order to obtain safe and efficient therapies against arterial calcification, future research should focus on developing novel routes of administration for these compounds thereby solely targeting the vessels and not the bone compartment.

## Figures and Tables

**Figure 1 ijms-21-07636-f001:**
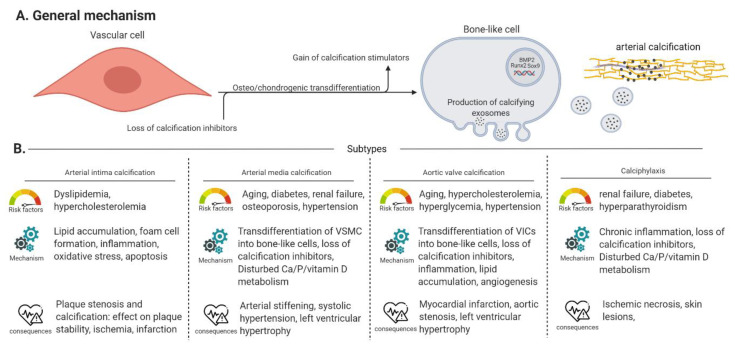
General mechanism and subtypes of arterial calcification. (Panel **A**) Describes the general mechanisms of arterial calcification. (Panel **B**) An overview of the risk factors, specific mechanisms and clinical consequences of arterial intima/media calcification, aortic valve calcification and calciphylaxis. VSMC: vascular smooth muscle cell, VIC: valve interstitial cell, Vascular cell: covers VSMCs, VICs and endothelial cells.

**Figure 2 ijms-21-07636-f002:**
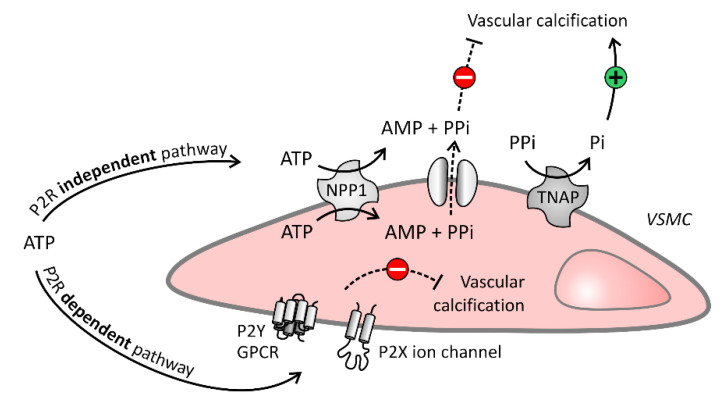
Purinergic independent and dependent signaling in the vascular cell. Purinergic receptor (P2R) independent pathway: Vascular cells degrade ATP into adenosine monophosphate (AMP) and pyrophosphate (PP_i_) by ecto-nucleotide pyrophosphatase/phosphodiesterase (NPP1). The potent calcification inhibitor PPi blocks arterial calcification, however, PP_i_ itself can also be broken down into the calcification stimulator inorganic phosphate (P_i_) by tissue non-specific alkaline phosphatase (TNAP). P2R dependent pathway: ATP also inhibits arterial calcification by interaction with P2 receptors, being P2Y and P2X receptors, expressed on the extracellular membrane of vascular cells. Solid and dashed arrows represent respectively stimulatory and inhibitory arrows.

**Figure 3 ijms-21-07636-f003:**
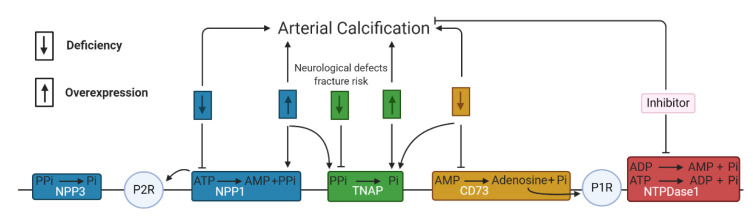
The reciprocal interaction between the different ecto-nucleotidases. The downward and upward arrow represents, respectively, a deficiency and overexpression of the ecto-nucleotidase activity. (**Blue**) NPP1 and NPP3 are involved in the synthesis and hydrolysis of pyrophosphate (PP_i_). Either a deficiency or overexpression of NPP1 activity leads to arterial calcification, as well as overexpression of NPP1 favoring TNAP activity and inducing less ATP mediated purinergic receptor 2 signaling (P2R). (**Green**) A deficiency of TNAP activity is linked to neurological defects and fracture risk by decreased P_i_ generation, while overexpression of TNAP activity favors arterial calcification through a reduction in PP_i_. (**Orange**) A deficiency of CD73 or 5′-nucleotidase triggers arterial calcification by promoting TNAP activity and reducing the production of adenosine and thus less adenosine mediated P1R signaling. (**Red**) Inhibition of NTPDase 1 reduces arterial calcification by diminishing P_i_ generation. This scheme is based on the results in multiple cell types including VSMCs, VICs and endothelial cells but also i.e., primary dermal fibroblasts from arterial calcifications due to deficiency of 5′-nucleotidase/CD73 (ACDC) patients.

**Figure 4 ijms-21-07636-f004:**
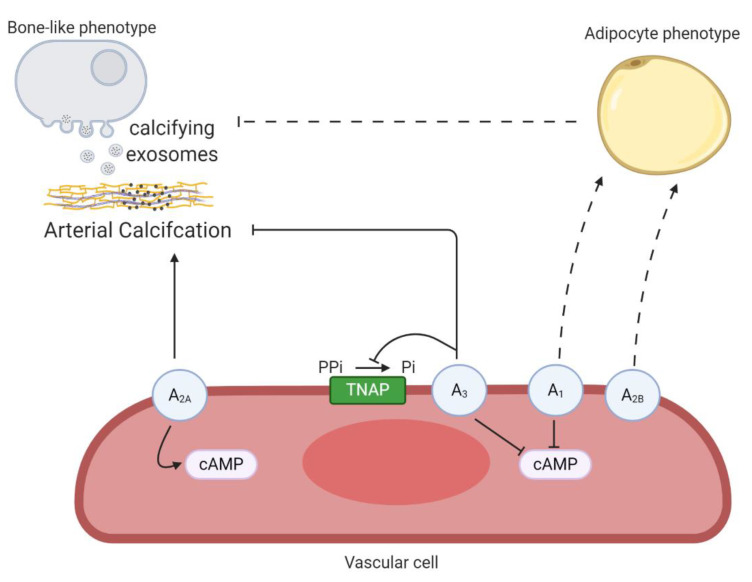
An overview the involvement of P1 receptors in arterial calcification. Activation of the A_2A_ receptor favors arterial calcification by stimulating intracellular cyclic adenosine monophosphate (cAMP) production. The A_3_ receptor blocks arterial calcification through decreasing tissue non-specific alkaline phosphatase (TNAP) activity and cAMP production. Both A_1_ and A_2B_ receptors might favor a phenotypic switch of vascular cells into an adipocyte phenotype and by this prevent the transdifferentiation towards a bone-like phenotype, thus halting the arterial calcification process. Dashed lines represent relevant findings in other cell-types, not yet shown in vascular cells. The term vascular cell covers vascular smooth muscle cells and valve interstitial cells.

**Figure 5 ijms-21-07636-f005:**
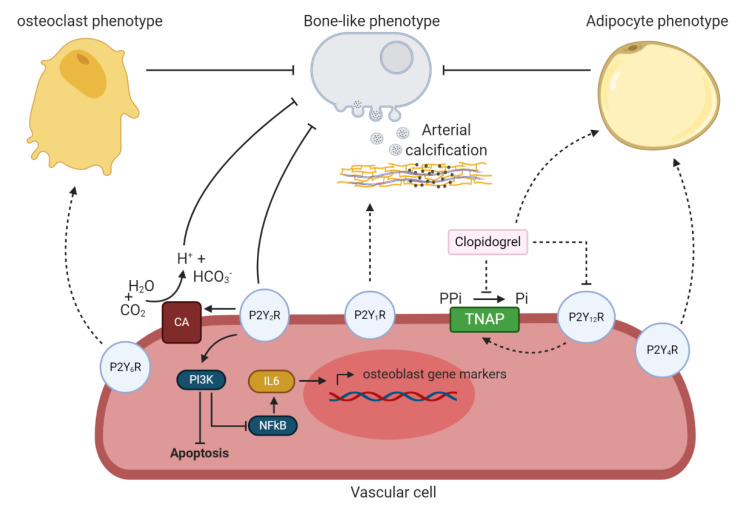
An overview the potential involvement of P2Y receptors in arterial calcification. P2Y_6_ receptor might trigger a phenotypic switch of vascular cells towards an osteoclast phenotype and by this prevent arterial calcification. P2Y_2_ receptor blocks arterial calcification by (i) stimulating phosphoinositide 3-kinases (PI3K) resulting in a reduction in vascular cell apoptosis and halting the bone-like phenotypic switch of vascular cells via less NFkB signaling and IL6 production and (ii) favoring carbonic anhydrase (CA) expression, catalyzing the reaction of water (H2O) and carbon dioxide (CO_2_) into hydrogen carbonate (HCO_3_^−^) and hydrogen ions (H^+^) leading to extracellular acidification and thus, resorption of the calcium-phosphate crystals. The P2Y_1_ receptor might stimulate the arterial calcification process. P2Y_12_ receptor blockage by clopidogrel potentially halts arterial calcification trough inhibition of tissue non-specific alkaline phosphatase (TNAP) and favoring the transdifferentiation of the vascular cell towards an adipocyte phenotype, thus preventing a bone-like phenotypic switch. P2Y_4_ receptor might also be involved in the phenotypic switch of vascular cells to an adipocyte phenotype. Dashed lines represent relevant findings in other cell-types, not yet shown in vascular cells. The term vascular cell covers vascular smooth muscle cells and valve interstitial cells.

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
