# Peer review of "Extracellular Nucleotides Regulate Arterial Calcification by Activating Both Independent and Dependent Purinergic Receptor Signaling Pathways"

_ijms, 2020, doi:10.3390/ijms21207636_

Round 1
Reviewer 1 Report
The current manuscript by Opdebeeck et al is a comprehensive review of literature on extracellular nucleotides, and their impact on cardiovascular calcification. This review also reveals some of the major implications of purinergic signaling that regulate cardiovascular calcification. All in all, this review gives a wholistic insight of the important role both extracellular nucleotide and purinergic signaling have on the progression of calcification and future considerations of possible therapies. I have no comments or corrections to give and congratulate the authors on a well written manuscript.
Author Response
I am very pleased to hear that the reviewer feels that my manuscript contributes to the important role of both extracellular nucleotides and purinergic signaling on the progression of calcification and future considerations of possible therapies.
Reviewer 2 Report
Overall a useful review article which should make a good contribution to an area which has generated many questions but few answers.
1. A frequent issue is the use of acronyms throughout the paper which are not defined at first usage.
2.Lines 42, 43: this sentence is confused and the poor English prevents it being understood.
3. I think the paper would benefit from a little more explanation as to the purposes and differences between extracellular ATP, UTP, CTP and GTP
4. Line 49 onwards: there can also be considerable calcification of atherosclerotic plaque, known as sub-clinical atherosclerosis, which you have not mentioned here.
5. Line 52: under the title of Arterial Calcification you include the aortic valve. I suggest changing the title to Arterial and Valvular Calcification. You will also then need to consider mitral annulus calcification, which you have not mentioned at all.
6. Line 58: you have not mentioned vitamin K as an important calcification inhibitor. Very simplistically, vitamin K is known to keep calcium out of the arteries and deposit it in bone. If you have not come across it before, take a look at Nicoll R et al in International Cardiovascular Forum, 2015.
7. Line 61: in the intimal layer as well.
8. Line 88 onwards: An explanation of purinergic receptors and the triphosphate of ATP would be helpful.
9. Line 111 onwards: Please explain the relationship between NPP1 the ENPP1 gene. This sentence would seem to suggest that the absence of the gene induced calcification, yet lower down you talk about an both overexpression of NPP1 and NPP1 inhibition being linked to calcification. If this is the case you need to make this clear and either suggest why or state that no-one knows why.
10. Line 126: 'mainly' twice in the same sentence. Need more detail on NPP3 and what is going on in allergy.
11. Line 172 onward: This suggests that TNAP inhibitors decrease both arterial calcifiction and bone formation. This is in contrast to vitamin K. Need to explain why.
12. Line 195: You give an example in valves and talk about arterial calcification.
13. Line 189-294: This whole paragraph is very confusion and needs some clarification at the end. You need to explain the relationship of NTPDases and TNAP if you are going to talk about them both in the same paragraph.
14. In general, I would suggest a 1 sentence summary at the end of each paragraph.
15. Line 270: Need to explain more about why PPARgamma inhibits calcification.
16. Line 300: Again, you should bring in vitamin K.
17. Line 306: And intimal calcification, not just medial.
18. Line 313: Need more explanation of the effect of fluid shear stress and extracellular calcium as this is the first mention of both. Also need a reference.
19. Line 380: need some explanation of why magnesium might inhibit arterial calcification.
Reviewer 3 Report
This manuscript by Opdebeeck et al. reviews on importance of extracellular nucleotides on the initiation and development of arterial calcification. The authors clearly review purinoceptor-dependent and -independent pathways on regulation of arterial calcification induced by extracellular nucleotides. Overall, this is nice review. I have some minor comments
- Firstly, the manuscript, as a whole, the abbreviation needs to be checked again. Please revise subscript in receptor subtypes and Up4A.
- I understand that VSMC is an important to development of arterial calcification, but endothelial cells are also important partly discussed by the authors. So, the authors should discuss about not only VSMCs but also ECs in some sections.
- In Figure 1A, please check again about “Vascular cell” that means ECs and VSMCs. If it means only VSMCs, please revise it.
- The authors described about uridine adenosine tetraphosphate, however, how about other dinucleotides? Please discuss them.
- If the authors used “significantly” in statistical means, please show comparison group, thoroughly.
- In line172, please show chemical name of SBI-425.
- In Figure 3, is this scheme in VSMC, EC, or both?
- In line 249, please revise from Patel JJ to Patel.
- As the authors reviewed, the balance among purinoceptors is very important to regulate arterial calcification. Are there any evidences about the alterations of these receptor subtypes expression in not only VSMCs but also ECs in some diseases related to arterial calcification?
- In lines 294~, please show detailed treatment regimen in the in vivo study.
- In line 314 release of ATP and UTP, what’s source? ECs?
- In Figure 5 vascular cell, is it VSMC only?
- In line 374, please describe name of antagonists.
Round 2
Reviewer 2 Report
All good. Thank you.